# Can Physical, Psychological, and Social Vulnerabilities Predict Ageism?

**DOI:** 10.3390/ijerph20010171

**Published:** 2022-12-22

**Authors:** Lea Zanbar, Sagit Lev, Yifat Faran

**Affiliations:** 1School of Social Work, Ariel University, Ariel 40700, Israel; 2Faculty of Social Work, Ashkelon Academic College, Ashkelon 78211, Israel; 3School of Social Work, Bar Ilan University, Ramat-Gan 52900, Israel; 4Department of Special Education, Hemdat Hadarom College Sdot Hanegev Regional Council, Netivot 8771302, Israel; 5Department of Gerontology, Ben Gurion University, Be’er Sheva 8410501, Israel

**Keywords:** ageism, vulnerabilities, well-being, post-traumatic distress, social support

## Abstract

Ageism can be expressed as the discrimination, social exclusion, and even abuse of older adults. The literature suggests that certain vulnerabilities could be risk factors affecting people’s ageism. Based on the Social Identity Theory, the present study aimed to examine the association of physical/psychological and social vulnerabilities with ageism. The sample consisted of 200 Israelis from the general population who completed self-report questionnaires. Hierarchical regression indicated that low well-being, high post-traumatic distress, and limited social support were associated with ageism. Furthermore, the association of post-traumatic distress with ageism increased with age. The findings expand the knowledge of vulnerabilities as risk factors for ageism, perhaps reflecting its unconscious nature, and can assist in designing interventions for people interacting with older adults.

## 1. Introduction

Ageism is defined as “negative or positive stereotypes, prejudice and/or discrimination against (or to the advantage of) elderly people on the basis of their chronological age or on the basis of a perception of them as being “old” or “elderly”. Ageism can be implicit or explicit and can be expressed on a micro-meso- or macro-level” [1] (p. 15).

Ageist behavior in the general population can result in social exclusion and social rejection, arousing feelings of loneliness among older adults [2]. In mental and physical health care, clinicians’ ageist stereotypes and prejudices might affect their assessment and treatment of older adults and can be reflected in inequality in the quantity and quality of health care compared to younger age groups [3,4]. This might result in the aggravation of older adults’ health conditions, a situation that can also have considerable economic implications for society at large [5].

In view of these harmful consequences, both practitioners and researchers have attempted to identify the risk factors for developing ageism in order to promote efforts to reduce it. The current study is therefore aimed at examining additional factors that might be associated with ageism.

Researchers have proposed several explanations regarding the factors underlying ageism, all of them derived from the common conception that ageism results from unconscious mechanisms of fear and a need to reduce stress [6]. Thus, for example, it is argued that ageism is triggered by the physical changes that characterize old age, which might be perceived as signs of disease. These signs unconsciously signal people to avoid contact with anyone who demonstrates them [7,8]. This explanation is supported by studies reporting an association between an increased perception of susceptibility to infections and ageism [9,10].

It may be assumed that such unconscious associations between old age and disease have been heightened by the COVID-19 pandemic, which has particularly affected older adults in terms of both mortality and the severity of physical and mental illness [11,12,13]. Consequently, people may now view older adults as even more vulnerable, increasing their need to differentiate themselves from the older population as a defense mechanism aimed at reducing the stress and fear of infection and thereby accentuating their ageism [14,15,16].

Hence, ageism serves as a defense mechanism used to differentiate ourselves from old age due to the inherent unconscious threats of disease and death [9,10]. Thus, it may be assumed that people who feel more vulnerable will also feel more threatened and, as a result, have a higher motivation to differentiate themselves from older adults. Indeed, it has been found that an increased perception of vulnerability to infections increases ageism [9,10]. These findings raise the question of whether the same mechanism operates in the presence of other vulnerabilities as well. In other words, do people characterized by vulnerabilities in other aspects of life, aside from illness or infectious diseases, display ageism in an unconscious attempt to not see themselves as suffering from the same vulnerabilities as older people, and thus avoid facing the fears this self-image arouses? Recent studies indeed demonstrate that further factors, specifically vulnerabilities related to aging, can serve as a trigger for ageism [10,17].

The current study is an exploratory investigation of additional factors that are identified in the literature as reflecting substantial vulnerabilities in life, and may consequently be risk factors for ageism. More specifically, we examined the association of ageism with factors indicating physical/psychological vulnerabilities (somatization, post-traumatic distress, and psychological well-being) and social vulnerabilities (social support and community belonging). The Social Identity Theory [18,19] might help explain why such vulnerabilities could lead to ageism, as it relates to the unconscious mechanism that urges people to differentiate themselves from others. We therefore chose this theory as the theoretical framework for our study.

### 1.1. The Social Identity Theory

According to the Social Identity Theory [18], a person’s identity is related to their membership in reference groups, as well as to their relationships with members of other groups. Consequently, in order to achieve a positive self-identity, individuals might demonstrate a bias against other groups so as to elevate their in-group status above that of others [18,20].

Thus, young and middle-aged people might attempt to differentiate themselves from, and elevate themselves above, older people in order to create a positive identity for their age group. This tendency might be heightened by society’s negative view of older adults [20]. The need for a positive distinction from the old age group may be even greater among middle-aged adults, who are aware that they are closer to becoming members of a devalued group [20]. This proposition is supported by findings showing that middle-aged participants (aged 40–67) are more ageist than younger and older groups [21].

### 1.2. Vulnerability in Old Age

Vulnerability is defined as “being at increased risk of harm, and/or having a decreased capacity to protect oneself from harm” [22] (p. 12). Older adults tend to be perceived as one of the most vulnerable populations, both by researchers [23] and by the general public [24,25]. Vulnerability in old age relates not only to the deterioration of physical and cognitive abilities [26] but also to sociological features, such as social networks and social power [27]. Furthermore, it is not merely associated with the personal characteristics of older adults, but also with the social environment, as older people tend to encounter discrimination, marginalization, and the loss of power and autonomy [23].

The relationship between old age and vulnerability is also reflected in stereotypes of older people. According to the Stereotype Content Model [28], older adults, similar to the intellectually and physically disabled, are perceived as incompetent. In some cases, however, this perception can lead to emotions of pity and sympathy as well [28].

Similarly, it may be assumed that individuals in vulnerable life situations might unconsciously associate their experience with the aging process and the vulnerabilities that accompany it. This unconscious association might intensify their anxiety about their own aging [29], increasing their ageism in their desire to differentiate themselves from older adults. Although to the best of our knowledge, the relationship between vulnerability and ageism has not previously been examined, a cumulative body of research indicates relationships between vulnerabilities, reflected in mental health conditions and substance abuse, and elder abuse [30,31].

While ageism is also related to social structures and social power on the macro level [23,27], the present study aims to examine its dynamics on the individual level. Expanding the examination of this dynamic is of utmost importance, since if vulnerability is indeed a risk factor for ageism, it should be identified and addressed among the people in the older adult’s surroundings, such as family members and caregivers. Moreover, most people can be expected to interact with an older adult at least at some point in their lives, and perhaps to be involved in their care, even if they are not currently.

### 1.3. Vulnerable Life Situations

In view of the connection between old age and vulnerability, the present study sought to examine whether ageism is displayed by people in major vulnerable life situations, who, according to the Social Identity Theory, may wish to distinguish themselves from older adults if they recognize the vulnerabilities associated with aging in their own lives [20,29]. This notion guided the selection of the variables in the current study, which represent vulnerabilities in main life domains: physical, psychological, and social. In each one of these domains, we chose factors that are identified in the literature as potential vulnerabilities in old age: the physical and psychological disorders of somatization, post-traumatic distress, (reduced) psychological well-being [32,33,34], and the social vulnerabilities of (reduced) social support and community belonging [35,36,37].

*Somatization* refers to somatic symptoms, such as headaches, back and muscle pain, and feeling tired, which are usually attributed to a mental state caused by adversity and stressful situations [38]. Empirical studies from Europe and Africa indicate a relatively high prevalence of somatization among older adults, pointing to rates of 25.3% and even 47% [39,40]. In addition, a comparison between young-old (60–70) and older-old (71+) found that the former tended to report somatic symptoms at a rate of 55% less than the latter (based on the “odds ratio” method of calculation) [40]. Moreover, since the prevalence of multimorbidity increases substantially with age [41], older adults are perceived to experience disease and pain as a “natural” part of the aging process [42].

*Post-traumatic distress* refers to a variety of symptoms stemming from a traumatic experience, including depression and anxiety [43]. According to DSM-5, post-traumatic stress disorder has severe negative implications on emotional, cognitive, and functional aspects of life [44]. Moreover, studies indicate that post-traumatic distress may be related to physical illness [45], poor mental health [46], living in economically distressed city neighborhoods [47], and limited social support [48]. Thus, people suffering from distress of this sort can be considered vulnerable. Studies suggest that PTSD symptoms become aggravated over time, particularly in the context of aging [49].

*Psychological well-being* relates to an individual’s cognitive and affective assessments of their lives [50]. Reduced well-being may derive from a range of diverse conditions, including job loss [51], health and physical deterioration [52], and living in economically distressed city neighborhoods among adolescents [47]. Moreover, food insecurity, unmet health care needs, increased morbidity, and reduced physiological flexibility among older adults adversely affect their well-being and thus increase their perceived vulnerability [33,52].

Findings regarding psychological well-being are mixed. While many studies indicate an improvement in mental health in old age, e.g., [53], others suggest that older adults’ well-being might be reduced by the risk factors that characterize old age, such as deterioration in cognitive and physical functioning, as well as by the lack of a partner, employment, and parental role identities [52,54].

*Social support* relates to both social and psychological forms of support that are received from non-professionals in the individual’s environment [55]. Studies indicate that the social networks of older adults tend to become more limited over time, thus putting them at greater risk of loneliness [27,56]. As older adults are perceived to have less social support, they are considered to be more vulnerable to other adversities in their lives [36,56].

*Community belonging* refers to the perception of being part of a collective that serves as a source of security and support [57]. Similar to social support, the lack of a sense of a supportive community can be experienced as a vulnerability. Indeed, populations with low social support and a lack of community belonging who were exposed to continuous rocket fire were found to be highly vulnerable [58], and social exclusion was shown to create vulnerability among Romanian beggars in Sweden [59]. A sense of community belonging may also diminish in old age. Older adults often feel they are no longer connected to the local community in which they have spent many years, or that their existing communities no longer meet their needs and lifestyle aspirations [60]. Moreover, older adults tend to prefer to stay at home, sometimes due to a lack of neighborhood security and safety, and are therefore likely to be less socially active and connected. The resulting low level of social functioning fosters the development of their frailty; that is, it increases their vulnerability [61].

### 1.4. The Present Study

As noted above, the present study aimed to examine whether physical/psychological and social vulnerabilities can predict ageism. We chose to employ a sample drawn from the general population for two reasons. First, every member of the population can be expected to interact with older adults, whether in a public, professional, or family setting. Secondly, this enabled us to recruit as diverse a sample as possible, including people of different ages, occupations, and so on. Based on Social Identity Theory and the literature on the link between certain vulnerabilities perceived to be related to old age and ageism, we hypothesized that individuals displaying less psychological well-being, social support, and community belonging, and more somatization and post-traumatic symptoms, would be more likely to display ageism. In addition, due to the central role of age in increasing ageism, we examined its direct associations with ageism, as well as the moderated associations with ageism that might result from its interaction with other vulnerability factors.

## 2. Materials and Methods

### 2.1. Participants

The sample size was calculated by means of G*Power (3.1.0 software) for 11 predictors, of which 5 are being tested, and a power of 0.9. For this condition, G*Power indicated that 187 participants were needed. In addition, we followed the common specific recommendation for regression analyses in the literature of N ≥ 50 + 8 m (m = number of predictors) when focusing on R^2^, or N ≥ 104 + m [62,63] when focusing on betas. Our sample therefore consisted of 200 Israelis from the general population, thereby verifying that reasonable statistical power can be achieved without obtaining significant effects that stem from too large a sample size or a needlessly overpowered design [64]. The age of the participants ranged from 19 to 76 years (M = 36.21, SD = 11.42). In terms of gender, 75.5% (n = 151) were women and 24.5% (n = 49) were men. Years of education ranged from 6 to 25 (M = 15.98, SD = 3.05). One hundred and one (50.5%) participants reported a lower-than-average monthly income, 33 (16.5%) reported an average income, and 66 (33%) stated that their monthly income was higher than average. The large majority (n = 146; 73%) were married or in a stable relationship, with only 54 (27%) indicating that they did not live with a partner.

### 2.2. Procedure

The study was approved by an academic ethics committee and was conducted in accordance with the ethical criteria for social research [65]. The questionnaire was distributed online via social networks. The participants were asked to complete self-report questionnaires, a task that required approximately 10 min. The anonymity and confidentiality of the participants were assured, and it was stressed that participation was voluntary and that there would be no consequences if they chose not to complete the questionnaire in full. All the participants gave their written informed consent to take part in the study. Due to the method of distribution, there was no way to determine the response rate. However, no comments indicating refusal to participate or difficulty completing the questionnaire were received.

### 2.3. Instruments

*The Fraboni Scale of Ageism (FSA)* [66] was used to assess explicit ageism. The instrument consisted of 29 statements regarding older adults and their place in society and related to three aspects of ageism—antilocution (stereotypes), avoidance, and discrimination (e.g., “*Many old people are stingy and hoard their money and possessions*”*;* “*I sometimes avoid eye contact with old people when I see them*”*;* “*It is best that old people live where they won’t bother anyone*”)—that can be expressed in both attitudes and behaviors. Participants were asked to indicate the degree to which they agreed with each of the items on a 4-point Likert scale ranging from 1 (*strongly disagree*) to 4 (*strongly agree*). Despite the instrument’s theoretical division into three aspects of ageism, factor analysis in the current study recognized only one factor for all 29 items, as was also the case in previous studies [67]. Accordingly, each participant was assigned a score equal to the mean of their responses to all items, with higher scores indicating stronger ageism. Cronbach’s alpha for the original questionnaire was 0.86 and was 0.84 in the current study.

*The Somatic Symptom Severity Scale (PHQ-15)* [38] was used to measure somatization. The original instrument consists of 15 items relating to various somatic symptoms that account for more than 90% of the physical complaints reported by outpatient settings and the most prevalent somatization disorder symptoms according to the DSM-5 (e.g., *stomach pain, dizziness*). Following previous studies [68], two items not relevant for males (menstrual cramps and problems during sexual intercourse) were omitted. Participants were asked how often they had been bothered by each of the symptoms during the last four weeks, indicating their answers on a 3-point Likert scale ranging from 1 (*not bothered at all*) to 3 (*bothered a lot*). Each participant was assigned a score equal to the sum of their responses, with higher scores indicating more symptoms of somatization. Cronbach’s alpha for the original questionnaire was 0.80 and was 0.78 in the current study.

*The PTSD Checklist for DSM-5 (PCL-5)* [43] was used to assess post-traumatic distress. Participants were asked how often they had been bothered by each of the 20 symptoms characteristic of PTSD during the previous month, with responses indicated on a 5-point Likert scale ranging from 1 (*not at all*) to 5 (*extremely*). Each participant was assigned a score equal to the sum of their responses, with higher scores indicating greater post-traumatic distress. A Cronbach’s alpha of 0.95 was reported in previous studies [69] and was the same in the current study.

*The Bradburn Scale of Psychologic Well-being* [70] was used to assess psychological well-being. Participants were asked whether they had recently experienced any of 10 emotions (e.g., “*Pleased about having accomplished something*”), marking their responses on a 4-point Likert scale ranging from 1 (*very often*) to 4 (*not at all*). Each participant was assigned a score equal to the mean of their responses, with higher scores indicating greater well-being. Cronbach’s alpha for the original scale ranged from 0.67 to 0.86 and was 0.75 in the current study.

*The Multidimensional Scale of Perceived Social Support (MSPSS)* [71] was used to assess social support. The 12-item scale measures perceived support from three sources: family, friends, and a significant other (e.g., “*I have a special person who is a real source of comfort to me*”). Participants were asked to indicate the degree to which they agreed with each of the items on a 5-point Likert scale ranging from 1 (*very strongly disagree*) to 5 (*very strongly agree*). Each participant was assigned a score equal to the mean of their responses, with higher scores indicating higher perceived social support. Cronbach’s alpha was 0.88 for the original questionnaire and 0.94 for the current study.

*The Psychological Sense of Community Index* [72] was used to measure the sense of community belonging. The questionnaire was originally developed to evaluate neighborhood cohesion and consisted of 18 statements regarding feelings about the individual’s community and neighborhood (e.g., “*I feel I belong to my community*”). Since some of the items applied to urban neighborhoods and some referred mainly to rural communities, in the current study we used a shortened version of the instrument, consisting of 11 items that are relevant for residents of both types of communities [73]. Participants were asked to indicate the degree to which they agreed with the statement in each item on a 5-point Likert scale ranging from 1 (*strongly disagree*) to 5 (*strongly agree*). Each participant was assigned a score equal to the mean of their responses, with higher scores indicating a stronger sense of community belonging. Cronbach’s alpha was 0.88 for the original scale and 0.95 for the current study.

A sociodemographic questionnaire was used to tap background variables, including age, gender, years of education, monthly personal income, and marital status (married or in a stable relationship vs. never married, widowed, separated, or divorced).

### 2.4. Data Analysis

Pearson correlations were conducted to examine the relationships between the background variables (some of which were entered as covariates), physical/psychological and social variables, and ageism. This was done to verify that the independent variables were all associated with the dependent variable of ageism, as well as to rule out the possibility of multicollinearity.

In the next stage, hierarchical regression analysis was performed to examine the unique and combined contribution of the independent variables to the explanation of the variance in ageism. The background variables (age, gender, years of education, income, marital status) were entered in the first step, the physical/psychological variables (somatization, post-traumatic distress, psychological well-being) in the second step, and the social variables (social support, community belonging) in the third step. In view of the literature indicating the substantial role of age in predicting ageism, its effect on the relationships between the other vulnerability variables and ageism was examined in the last step of the regression. Since the literature does not indicate any specific interaction, all of the possible interactions between age and each of the physical/psychological and social variables were examined (i.e., five possible interactions) using the stepwise method, which presents only the significant results.

Finally, in order to determine the nature of the effect of age on the associations of the vulnerability variables with ageism, the single interaction found significant in the regression analysis was further examined by model no. 1 of PROCESS, which tests for moderation relationships between the variables [74]. The constituent variables were centered before the interaction terms were computed as their cross-products [75].

## 3. Results

Table 1 presents the intercorrelations among all study variables and their descriptive statistics (means and standard deviations for dichotomous variables were omitted; their frequencies are presented in the text).

The results of the regression analysis, performed to examine the unique and combined contribution of the independent variables to the explanation of the variance in ageism, appear in Table 2.

As Table 2 indicates, all ten predictor variables and one interaction explained 23% of the variance in ageism, with adjusted *R^2^* = 0.18, *F*(11, 188) = 4.96, *p* < 0.001. Furthermore, an examination of the variance inflation factor (VIF) values (all close to 1) confirmed that the results of the regression analysis were not affected by multicollinearity.

The background variables entered in Step 1 accounted for 7% of the variance in ageism; *F*(5, 194) = 3.05, *p* < 0.05. Older participants reported stronger ageism, while participants with higher education reported lower ageism. Gender, income, and marital status were not associated with ageism. The physical/psychological variables in Step 2 explained an additional 8% of the variance in ageism, *F*(3, 191) = 5.59, *p* < 0.01. Participants reporting higher levels of post-traumatic distress also displayed stronger ageism, and those reporting higher levels of psychological well-being exhibited lower levels of ageism. Somatization was not associated with ageism. The social variables in Step 3 explained 1.5% of the variance, with non-significant values of variance of *F*(2, 189) = 1.53, *p* > 0.05. Only social support had a significant effect on ageism, indicating an association between higher social support and lower levels of ageism.

Finally, of the five possible interactions tested in Step 4, only the interaction between age and post-traumatic distress reached significance, explaining a further 6.5% of the variance; *F*(1, 188) = 15.42, *p* < 0.001. Following the recommendation of Cohen and Cohen (1983) [76], this interaction was plotted based on values corresponding to the mean, one standard deviation above the mean, and one standard deviation below the mean for age and post-traumatic distress. In addition, a simple slope analysis was conducted using the PROCESS procedure for SPSS (Release 2.00) [74]. Figure 1 presents the plots of the effect of the interaction between age and post-traumatic distress on the prediction of ageism (*t* = 2.95, *p* < 0.01).

As can be seen from Figure 1, the effect of post-traumatic distress on ageism increased with age. Among younger participants, no significant correlation emerged between the two variables. Accordingly, the simple slope for one standard deviation below average was *β* = 0.007, *SE* = 0.01, *p* = 0.31. The slope for participants of average age was *β* = 0.03, *SE* = 0.01, *p* < 0.001, and for participants with age one standard deviation above average it was *β* = 0.04, *SE* = 0.01, *p* < 0.001. Thus, the effect of PTSD on ageism was more pronounced the older the participants.

## 4. Discussion

The current study examined the association of physical/psychological and social vulnerabilities with ageism. In line with our hypothesis, all the study variables, save for somatization and community belonging, were found to be associated with ageism. Our findings, therefore, suggest that ageism can indeed be explained by an unconscious attribution of psychological and social vulnerabilities to old age. In other words, people might display ageism in order to differentiate themselves from the vulnerability associated with old age, particularly if they perceive themselves to be similarly vulnerable.

Thus, the present study contributes to the theoretical and empirical literature concerning the unconscious nature of ageism [6,29]. It suggests that a subjective experience of vulnerability might unconsciously be associated with negative aspects of the aging process, thereby intensifying individuals’ anxiety about their own aging and resulting in ageism aimed at differentiating themselves from older adults, in line with the Social Identity Theory. This mechanism can also be understood in light of similar unconscious mechanisms described by the Disease Avoidance Theory [8]. However, due to the exploratory nature of the current study, further research is required to provide a deeper understanding of the link between vulnerability and ageism.

With respect to the physical/psychological variables, as predicted, high post-traumatic distress and low personal well-being were both found to be associated with ageism. Although somatization correlated significantly with ageism, the regression analysis showed that it did not contribute to the explanation of its variance. It is possible that since somatization also correlated with other variables that proved to be significantly associated with ageism, much of its potential contribution to the explained variance in ageism was absorbed by those variables.

In the social realm, as expected, low social support was found to be associated with ageism. Although community belonging correlated with ageism, it did not make a significant contribution to the explanation of the variance. This might stem from the complex relations between these two variables. According to the Social Identity Theory, people form a positive self-identity by creating a distinction between their group (the in-group) and others (out-groups) and perceiving the status of the in-group to be higher [18,20]. Thus, people with high community belonging might be more inclined to display ageism in order to distinguish their community from the group of older adults [77]. People with low community belonging might also display more ageism because of their greater vulnerability [17]. The fact that individuals with both high and low community belonging might tend to be more ageist, albeit for different reasons, could make it impossible for this variable to function as a predictor.

Two demographic variables, years of education and age, were also found to be associated with ageism. The negative association that emerged between education and ageism is consistent with a previous study indicating that more years of education is associated with lower ageism [78]. Although there is no indication that the education acquired contained contents relating specifically to ageism and/or efforts to reduce it, it may be assumed that higher education in general exposes people to content that promotes greater tolerance and respect for the various sectors of society at large, including those who are vulnerable and older adults.

With respect to age, older participants were found to display greater ageism. The age of the participants in the current study ranged from 19 to 76. Thus, the oldest participants in the sample belonged to the middle-aged group (40–67) and young-old-age groups (67–84) [21,79]. Ageism in these two age groups can again be explained by the unconscious tendency to differentiate themselves from the vulnerability associated with old age. Among middle-aged people, this tendency may be spurred by the recognition that they are approaching old age [20], while among the young-old it may be motivated by the desire to preserve their “young” status by creating a separation between the “third age,” which signifies the young-old, and the “fourth age,” which relates to the old-old (85+) and represents “real old age” [79].

Finally, the analysis of the interaction between post-traumatic distress and age showed that the positive association between post-traumatic distress and ageism increased with age and was not significant for young participants. This might be explained by the unique characteristics of post-traumatic distress in old age. Traumatic events can have a cumulative effect on older adults, increasing the risk of the breakdown of psychological defenses [80]. Consequently, older adults who suffer from post-traumatic distress may be more vulnerable than younger people. This greater vulnerability might increase the motivation of this group to differentiate themselves from the old age group, and as a result, intensify their ageism.

Several limitations of the current study should be noted. First, although the findings indicate that the subjective experience of vulnerability is associated with ageism, as this is an exploratory study, further research into this unconscious mechanism, whether in light of the Social Identity Theory or additional or alternative theories, is needed.

Secondly, the study was conducted at a single point in time and relied on self-report measures. Therefore, causality cannot be determined. As attitudes develop over time, and certain vulnerabilities may have a cumulative effect, a longitudinal study of the link between vulnerability and ageism is warranted both to confirm our findings and to provide further insight into the dynamics of this association. Thirdly, the sample contained more women than men, and the average years of education was relatively high. Therefore, the generalization of the findings to the general population should be made with caution, and future studies should attempt to reach a more representative sample (in the current study, data collection was conducted via social networks, which did not enable us to control these features). In addition, further studies should control for measures indicating the identity of the participants with regard to older adults, such as how similar they perceive themselves to be to older adults and to what extent they interact with older adults. Future studies might also examine the moderating role of intergenerational contact and educational interventions in the relationships between vulnerable life situations and ageism.

Thirdly, the current study’s use of a scale to examine explicit ageism [66] can lead to a social desirability bias [81]. Employing tools that examine implicit ageism as well could overcome this bias and therefore deepen and refine our knowledge of this phenomenon [82]).

Moreover, as noted above, the sample chosen for the study was drawn from the general population on the assumption that anyone may potentially interact closely with older adults and affect their well-being. Moreover, as this is the first study to examine the link between physical/psychological and social vulnerabilities and ageism, we wished to be able to generalize the findings as widely as possible. Therefore, the levels of the vulnerability measures were similar to those that characterize the general population, ranging in continuous levels, rather than indicating extreme levels of these vulnerabilities, and in some cases even demonstrating that only a few of the participants reported levels indicating vulnerability (e.g., the measures of PTSD and well-being). Nevertheless, the diversity in the sample appeared to be enough to enable the identification of significant associations between these vulnerabilities and ageism, indicating that higher levels are associated with higher levels of ageism. Thus, in view of the insights gained here regarding vulnerabilities that may be risk factors for ageism, future studies would do well to focus on specific populations, such as family members, professionals working with older adults, and caregivers. It might also be of value to examine the impact of the ethnicity of participants on their ageist attitudes.

Finally, future studies might also investigate whether the link between vulnerabilities and ageism revealed in the current study could be found with respect to other stereotypes, as well. It would be interesting to learn whether other stereotypes and prejudices toward additional social groups, including people with disabilities, minority groups, and so on, are associated with vulnerabilities.

## 5. Conclusions

The findings of the current study have both theoretical and practical implications. In line with other studies and theories [6,9], they reveal the unconscious nature of ageism, showing that we might recoil from older adults because of our unconscious fears of vulnerability, disease, and death, which are associated with the aging process.

Furthermore, due to the negative ramifications of ageism [83,84], the results of this study should be considered by professionals working with older adults and may aid in the design of professional interventions aimed at reducing ageism among the significant figures in their surroundings, particularly family members and caregivers [85,86]. The findings suggest that when these individuals are in a vulnerable state as a result of physical, psychological, or social circumstances, they might be more likely to display ageism. Indeed, research indicates that caregivers may be at high risk of economic, emotional, and physical vulnerability [87,88] and that vulnerabilities including poor physical and mental health increase the risk of elder abuse and neglect [17]. It is therefore of great importance for professionals to direct their attention to the people around older adults to ensure that any vulnerabilities they themselves may be dealing with are not reflected in unhealthy attitudes or behavior toward the older adults.

To conclude, this study indicates, for the first time, that beyond the adverse social, emotional, and health effects of low psychological well-being, high post-traumatic distress, and limited social support, these characteristics also put people at greater risk of developing ageism, thus passing forward their vulnerability. Efforts should therefore be made not only to assist such individuals in coping with their vulnerabilities but also to prevent their expansion to other potentially vulnerable populations, particularly older adults.

## Figures and Tables

**Figure 1 ijerph-20-00171-f001:**
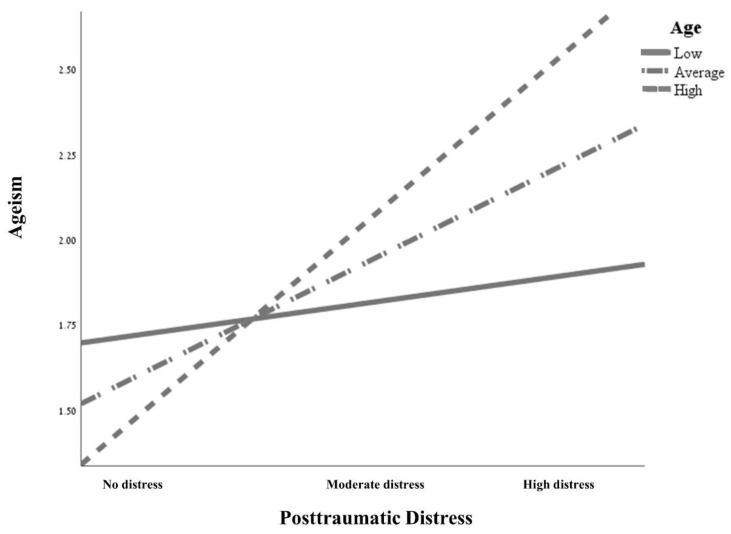
Interaction between Age and Posttraumatic Distress Predicting Ageism.

**Table 1 ijerph-20-00171-t001:** Intercorrelations among the study variables.

Measures	1	2	3	4	5	6	7	8	9	10	11
1. Age	----										
2. Gender	−0.19 **	----									
3. Years of education	0.43 ***	−0.02	----								
4. Income	0.47 ***	−0.19 **	0.40 ***	----							
5. Marital status	−0.24 ***	−0.05	−0.32 ***	−0.25 ***	----						
6. Somatization	−0.27 ***	0.31 ***	−0.22 **	−0.30 ***	0.26 ***	----					
7. Post-traumatic distress	−0.12 *	0.07	−0.25 ***	−0.09	0.33 ***	0.52 ***	----				
8. Psychological well-being	0.11	0.19 **	0.20 **	0.17 *	−0.33 ***	−0.35 ***	−0.53 ***	----			
9. Social support	0.04	0.13 *	0.09	0.02	−0.42 ***	−0.25 ***	−0.51 ***	0.61 ***	----		
10. Community belonging	0.23 **	0.10	0.22 **	0.19 **	−0.39 ***	−0.31 ***	−0.36 ***	0.50 ***	0.55 ***	----	
11. Ageism	0.01	0.03	−0.20 **	−0.15 *	0.15 *	0.22 **	0.30 ***	−0.27 ***	−0.27 ***	−0.15 *	----
*Mean*	36.21	----	15.98	----	----	19.73	36.92	3.01	4.12	3.30	1.78
*Standard Deviation*	11.42	----	3.05	----	----	4.45	16.23	0.48	0.85	0.98	0.35

* *p* < 0.05. ** *p* < 0.01. *** *p* < 0.001. As can be seen from Table 1, significant correlations were found between all the independent variables and ageism. In addition, no particularly high correlations between the independent variables were found that might indicate multicollinearity.

**Table 2 ijerph-20-00171-t002:** Hierarchical regression coefficients for the variance in ageism.

Variables	Coefficients						R-Square
	*Β*	B	SE	CI		VIF	
				Lower bound	Upper bound		
Step 1: *Background variables*							*R*^2^ = 0.073 *
Age	0.180 *	0.005	0.003	0.001	0.010	1.44	
Gender	0.042	0.034	0.058	−0.080	0.148	1.07	
Years of education	−0.189 *	−0.021	0.009	−0.040	−0.003	1.38	
Income	−0.126	−0.049	0.032	−0.111	0.014	1.41	
Marital status	00.100	0.078	0.058	−0.036	0.192	1.15	
Step 2: *Physical/psychological variables*							*R*^2^ = 0.148 ***, Δ*R*^2^ = 0.075
Somatization	0.052	0.004	0.007	−0.009	0.018	1.70	
Post-traumatic distress	0.168 *	0.004	0.002	0.000	0.007	1.82	
Psychological well-being	−0.142 *	−0.102	0.061	−0.223	0.019	1.64	
Step 3: *Social variables*							*R*^2^ = 0.161 ***, Δ*R*^2^ = 0.014
Social support	−0.170 *	−0.070	0.040	−0.148	0.009	2.14	
Community belonging	0.057	0.020	0.030	−0.040	0.080	1.68	
Step 4: *Interactions of age with independent variables*							*R*^2^ = 0.225 ***, Δ*R*^2^ = 0.064
Age X Post-traumatic distress	0.277 ***	0.001	0.000	0.001	0.002	---	

Note. The entries are values of *R^2^* for each step of the hierarchical regression and coefficients for each predictor, obtained when the variable was first entered. The values of the VIF (variance inflation factor) obtained in the last step included all the variables in order to rule out multicollinearity. * *p* < 0.05, *** *p* < 0.001.

## Data Availability

Data are available upon request. Please contact the authors.

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
