# Peer review of "Can Physical, Psychological, and Social Vulnerabilities Predict Ageism?"

_ijerph, 2022, doi:10.3390/ijerph20010171_

Round 1

Reviewer 1 Report (Previous Reviewer 1)

The authors have addressed my concerns adequately, and I have no further comments

Author Response

We would like to sincerely thank you for your comments, which helped us improve the quality of the paper.

Reviewer 2 Report (New Reviewer)

Dear authors,

This is an interesting study that examines the associations between physical, psychological and social vulnerabilities and individuals’ ageism and is well written overall. There are a few clarifications and edits that I suggest will improve the clarity and accuracy of your article, though I cannot comment on your data analysis methods as they are not my area of expertise.

·         Ageism also has health and  cost impacts of ageism, so this reference might be useful to justify the importance of your research and could be added to the references in line 31 - Becca R Levy, PhD, Martin D Slade, MPH, E-Shien Chang, MA, Sneha Kannoth, MPH, Shi-Yi Wang, MD, PhD, Ageism Amplifies Cost and Prevalence of Health Conditions, The Gerontologist, Volume 60, Issue 1, February 2020, Pages 174–181, https://doi.org/10.1093/geront/gny131

·         Though you talk about ageism throughout, it is important to clarify where ageism is mentioned throughout the manuscript whether you are talking about ageist thoughts, feelings or behaviours or all of these, as people who hold ageist attitudes might not necessarily behave in ageist ways. Ageism is nuanced with different dimensions (thoughts, feelings and behaviours), has different reference targets (i.e. self and others), and accounts for both positive and negative ageism, and explicit and implicit manifestations, as mentioned in Ayalon, L., Dolberg, P., Mikulionienė, S., Perek-Białas, J., Rapolienė, G., Stypinska, J., Willińska, M. &  de la Fuente-Núñez, V. (2019). A systematic review of existing ageism scales, Ageing Research Reviews, Volume 54, https://doi.org/10.1016/j.arr.2019.100919.

·         In your article you focus more on individual factors in developing ageist attitudes and behaviours. Ageism develops in a broader social context where social and cultural views of ageing and older people also impact. This is mentioned in passing, e.g. line 80-81 with reference 19 and in the next section – line 91-94 with reference 22, but I think it is important to highlight the impact of these social and cultural factors and distinguish that you are examining the link between ‘individual’ factors and ageism in your study. You talk about stereotyping and impacts of the social environment e.g. discrimination etc, but do not really explicitly link this with how individuals ageist attitudes can develop, as this is written in the ‘vulnerability in old age’ section.

·         The paragraph line 104-line 109 as written is an oversimplification and misrepresentation of the cited study [16] as you have not explained the relational link between the risk factors and vulnerabilities for both perpetrators and victims -  elder abuse victims also often have the same vulnerabilities and there is a relational element. Also, the study did consider physical, mental health and relational domains for the perpetrator, in conflict with your assertion in the paragraph. It is other studies cited within this one that talk about the association between negative attitudes toward older persons and elder abuse perpetration, so you would perhaps be better off citing one of those studies.

·         Line 334 – where you mention that older participants reported stronger ageism, is it possible to disaggregate whether this was thoughts, feelings or behaviours? This is important in line with my earlier comments.

·         Sentence in lines 408-409 which talks about the negative association between education and ageism. This point could be further discussed as it points to educative interventions having a role in reducing ageism, which could be included when you talk about interventions in your conclusion.

·         Line 445 – an important point made around future studies asking about to what extent participants interact with older adults ad it could be assumed that there is a relationship between experience and interaction with older adults and more positive views of them.

·         It is interesting that your sociodemographic questions did not ask about ethnic and cultural background. Perhaps there is not much variation of these in Israel, but in my context questions around these aspects would have been included in line with my earlier comments around the influence of the sociocultural context in developing vies and stereotypes of ageing. If this could be a factor in Israel, consider adding this to your limitations.

·         Also In limitations, you could also discuss the limitations of the measurement tool with reference to this study -   Ayalon, L., Dolberg, P., Mikulionienė, S., Perek-Białas, J., Rapolienė, G., Stypinska, J., Willińska, M. &  de la Fuente-Núñez, V. (2019). A systematic review of existing ageism scales, Ageing Research Reviews, Volume 54, https://doi.org/10.1016/j.arr.2019.100919.

Typographic and writing matters:

·         Line 132 - add the overall percentage figure here for younger adults to make it more easily comparable with the first two percentage figures provided in lines 130 and 131, then you can state that the overall rate is 55% less than that overall percentage for older adults

·         In several places you have used e.g., before a numbered reference – line 88, line 89, line 90, line 152, line 155, line 471. This would make more sense if the authors names were visible, so I suggest deleting the ‘e.g.’ as they are not necessary.

·         Add a reference for the ethical criteria you have used – line 207.

·         Line 223 – should be .82 rather than ,82

·         Sentence on lines 374-375 is not grammatically correct for the last 5 words – you need to include what is being attributed to old age.

·         Line 381 – suggest you add ‘negatively’ to associated, e.g. associated negatively

·         Line 431 – change ‘predicts’ to ‘is associated with’ as you say below that causality cannot be determined.

Author Response

Reviewer 2

Dear authors,

This is an interesting study that examines the associations between physical, psychological and social vulnerabilities and individuals’ ageism and is well written overall. There are a few clarifications and edits that I suggest will improve the clarity and accuracy of your article, though I cannot comment on your data analysis methods as they are not my area of expertise.

  1. Ageism also has health and cost impacts of ageism, so this reference might be useful to justify the importance of your research and could be added to the references in line 31 - Becca R Levy, PhD, Martin D Slade, MPH, E-Shien Chang, MA, Sneha Kannoth, MPH, Shi-Yi Wang, MD, PhD, Ageism Amplifies Cost and Prevalence of Health Conditions, The Gerontologist, Volume 60, Issue 1, February 2020, Pages 174–181, https://doi.org/10.1093/geront/gny131

R1. We thank you for the recommendation, and have added reference to this article in our literature review (Page 1, lines 32-34):

This might result in aggravation of the older adults’ health condition, a situation that can also have considerable economic implications for society at large [5].

  1. Though you talk about ageism throughout, it is important to clarify where ageism is mentioned throughout the manuscript whether you are talking about ageist thoughts, feelings or behaviours or all of these, as people who hold ageist attitudes might not necessarily behave in ageist ways. Ageism is nuanced with different dimensions (thoughts, feelings and behaviours), has different reference targets (i.e. self and others), and accounts for both positive and negative ageism, and explicit and implicit manifestations, as mentioned in Ayalon, L., Dolberg, P., Mikulionienė, S., Perek-Białas, J., Rapolienė, G., Stypinska, J., Willińska, M. & de la Fuente-Núñez, V. (2019). A systematic review of existing ageism scales, Ageing Research Reviews, Volume 54, https://doi.org/10.1016/j.arr.2019.100919.

R2. Thank you for your important comment. Following your suggestion, we have expanded the definition of ageism of Iversen et al. (2009) to include implicit and explicit aspects and micro, meso, or macro levels of ageism (Page 1, lines 24-27). As we saw that Ayalon et al.'s reference to ageism was based on Iversen's definition, we cited this reference directly:

Ageism is defined as "negative or positive stereotypes, prejudice and/or discrimination against (or to the advantage of) elderly people on the basis of their chronological age or on the basis of a perception of them as being ‘old’ or ‘elderly’ Ageism can be implicit or explicit and can be expressed on a micro-meso- or macro-level"

In addition, at the beginning of the Introduction section, we emphasize the behavioral dimension of ageism, while describing its negative consequences (Page 1, lines 29-34). However, in the rest of the literature review, we treat ageism as a broad term that includes diverse aspects, discussing its origins and its possible relationship with vulnerable life situations.

The broader perception of ageism is also manifested in the design of the empirical study. We used the Fraboni Scale of Ageism, which examines three dimensions of explicit ageism: antilocution (stereotypes), avoidance, and discrimination, which can be expressed both in attitudes and in behaviors. Although we provide examples of each of these dimensions, we note that the factor analysis we conducted on the scale found one global factor of ageism reflected in all 29 items, similar to previous studies. Accordingly, we computed a single score for the scale (Page 5, lines 217-228):

The Fraboni Scale of Ageism (FSA) [66] was used to assess explicit ageism. The instrument consists of 29 statements regarding older adults and their place in society, and relates to three aspects of ageism: antilocution (stereotypes), avoidance, and discrimination (e.g. Many old people are stingy and hoard their money and possessions; I sometimes avoid eye contact with old people when I see them;It is best that old people live where they won’t bother anyone”), that can be expressed in both attitudes and behaviors. Participants were asked to indicate the degree to which they agree with each of the items on a 4-point Likert scale ranging from 1 (strongly disagree) to 4 (strongly agree). Despite the instrument’s theoretical division into three aspects of ageism, factor analysis conducted in the current study recognized only one factor for all 29 items, as was also the case in previous studies [67]. Accordingly, each participant was assigned a score equal to the mean of their responses to all items, with higher scores indicating stronger ageism.

For additional studies using one score for all 29 items, please see, for example:

  • Rababa, M.; Hammouri, A.M.; Hweidi, I.M.; Ellis, J.L. Association of nurses' level of knowledge and attitudes to ageism toward older adults: Cross‐sectional study. Nursing & Health Sciences, 2020, 22(3), 593-601. https://doi.org/10.1111/nhs.12701
  • Yoelin, A.B. Intergenerational service learning within an aging course and its impact on undergraduate students’ attitudes about aging. Journal of Intergenerational Relationships, 2022,20(3), 277-292. https://doi.org/10.1080/15350770.2021.1881019
  • Lasher, K.P.; Faulkender, P. J. Measurement of aging anxiety: Development of the anxiety about aging scale. The International Journal of Aging and Human Development, 1993, 37(4), 247-259.‏ https://doi.org/10.2190%2F1U69-9AU2-V6LH-9Y1L

  1. In your article you focus more on individual factors in developing ageist attitudes and behaviours. Ageism develops in a broader social context where social and cultural views of ageing and older people also impact. This is mentioned in passing, e.g. line 80-81 with reference 19 and in the next section – line 91-94 with reference 22, but I think it is important to highlight the impact of these social and cultural factors and distinguish that you are examining the link between ‘individual’ factors and ageism in your study. You talk about stereotyping and impacts of the social environment e.g. discrimination etc, but do not really explicitly link this with how individuals ageist attitudes can develop, as this is written in the ‘vulnerability in old age’ section.

R3. Thank you for your important comment. We have now emphasized this point (Page 3, lines 109-110):

While ageism is also related to social structures and social power on the macro level [23,27], the present study aims to examine its dynamics on the individual level.

  1. The paragraph line 104-line 109 as written is an oversimplification and misrepresentation of the cited study [16] as you have not explained the relational link between the risk factors and vulnerabilities for both perpetrators and victims - elder abuse victims also often have the same vulnerabilities and there is a relational element. Also, the study did consider physical, mental health and relational domains for the perpetrator, in conflict with your assertion in the paragraph. It is other studies cited within this one that talk about the association between negative attitudes toward older persons and elder abuse perpetration, so you would perhaps be better off citing one of those studies.

R4. We thank you for this comment. The purpose of the paragraph was to indicate the possible relationships between vulnerable life situations and ageism. Since these relations have not been examined in previous studies, we offer examples of studies that indicate relationships between perpetrators' vulnerabilities, reflected in mental health issues and substance abuse, and elder abuse (Page 3, lines 105-108):

Although, to the best of our knowledge, the relationship between vulnerability and ageism has not previously been examined, a cumulative body of research indicates relationships between vulnerabilities, reflected in mental health conditions and substance abuse, and elder abuse [30,31].

  1. Line 334 – where you mention that older participants reported stronger ageism, is it possible to disaggregate whether this was thoughts, feelings or behaviours? This is important in line with my earlier comments.

R5. As mentioned in our response to comment no. 2, Fraboni's scale of ageism assesses both attitudes and behavioral tendencies (Page 5, lines 217-222). Moreover, following factor analysis and in line with numerous other studies, the score for each participant was calculated as the mean of all 29 items of the scale (rather than calculating three different scores for the components of the instrument). Hence, it cannot be determined which of the three dimensions of the scale impacted each of the findings. Rather, as consistent with our reference to ageism in general throughout the paper, we related to the scale as a whole.

  1. Sentence in lines 408-409 which talks about the negative association between education and ageism. This point could be further discussed as it points to educative interventions having a role in reducing ageism, which could be included when you talk about interventions in your conclusion.

R6. We have added the following possible explanation of the findings: (Page 11, lines 416-420)

Although there is no indication that the education acquired contained contents relating specifically to ageism and/or efforts to reduce it, it may be assumed that higher education in general exposes people to contents that promote greater tolerance and respect for the various sectors of society at large, including those who are vulnerable and older adults.

We have also added the following recommendation for future studies (Page 12, lines 455-457):

Future studies might also examine the moderating role of intergenerational contacts and educational interventions in the relationships between vulnerable life situations and ageism.

  1. Line 445 – an important point made around future studies asking about to what extent participants interact with older adults and it could be assumed that there is a relationship between experience and interaction with older adults and more positive views of them.

R7. We thank you for this comment. As noted in our response to comment 6, we have added the recommendation that future studies examine the possible contribution of intergenerational contacts in moderating the relationships between vulnerable life situations and ageism,

  1. It is interesting that your sociodemographic questions did not ask about ethnic and cultural background. Perhaps there is not much variation of these in Israel, but in my context questions around these aspects would have been included in line with my earlier comments around the influence of the sociocultural context in developing vies and stereotypes of ageing. If this could be a factor in Israel, consider adding this to your limitations.

R8. Thank you for this suggestion. Reference to ethnicity has been added to the limitations section (Page 12, lines 476-477):

It might also be of value to examine the impact of the ethnicity of participants on their ageist attitudes.

  1. Also in limitations, you could also discuss the limitations of the measurement tool with reference to this study - Ayalon, L., Dolberg, P., Mikulionienė, S., Perek-Białas, J., Rapolienė, G., Stypinska, J., Willińska, M. & de la Fuente-Núñez, V. (2019). A systematic review of existing ageism scales, Ageing Research Reviews, Volume 54, https://doi.org/10.1016/j.arr.2019.100919.

R9. We have added this limitation (Page 12, lines 458-461):

Thirdly, the current study's use of a scale to examine explicit ageism [66] can lead to a social desirability bias [81]. Employing tools that examine implicit ageism as well could overcome this bias and therefore deepen and refine our knowledge of this phenomenon [82].

Typographic and writing matters:

  1. Line 132 - add the overall percentage figure here for younger adults to make it more easily comparable with the first two percentage figures provided in lines 130 and 131, then you can state that the overall rate is 55% less than that overall percentage for older adults

R1. In the cited study (Mutepfa et al., 2021), the researchers calculated the differences between the two age groups’ reports on somatization by means of a statistical method called the "odds ratio," which computes one group’s odds of reaching the same prevalence of the measured parameter with the odds of the other group in relation to their size in the whole sample. In that study, the number of people reporting somatization in the young-old group was 30 out of 191 participants, that is, 15.7% of the group, whereas in the old-old group, it was 65 out of 185 participants, that is, 35.1% of the group. Based on these percentages, the researchers conducted the “odds ratio” calculation to examine the differences between the groups, treating the group of old-old adults as a whole, that is, as 100%, and testing the ratio of somatization of the second group, the young-old adults, in relation to it. Calculating the ratio between the above percentages of 15.7/35.1 in this way provided the researchers with a percentage of 45 of somatization for the young-old group in relation to the old-old group, which as explained was considered 100%, leading to a discrepancy of 55% between the rates of 100% and 45%. Consequently, they concluded that the difference between the rates of the prevalence of somatization in the groups was 55 percent.

For further information, please see Table 3 in the cited paper, and the explanation of the authors that follows it:

       Mutepfa, M. M.; Motsamai, T. B.; Wright, T. C.; Tapera, R.; Kenosi, L. I. Anxiety and somatization: Prevalence and correlates of mental health in older people (60+ years) in Botswana. Aging & Mental Health, 2021, 25(12), 2320-2329.

So as not to confuse the readers with a detailed and complex explanation and elaboration of statistical methods, values, etc., we did not include this explanation in our paper, but rather summarized the main findings. We have now added the comment that the calculation was based on “odds ratio,” enabling interested readers to view the full and detailed explanation in the paper of Mutepfa et al. (Page 3, lines 131-133):

In addition, a comparison between young old (60-70) and older old (71+) found that the former tended to report somatic symptoms at a rate of 55% less than the latter (based on the “odds ratio” method of calculation) [40].

  1. In several places you have used e.g., before a numbered reference – line 88, line 89, line 90, line 152, line 155, line 471. This would make more sense if the authors names were visible, so I suggest deleting the ‘e.g.’ as they are not necessary.

R2. Thank you for your suggestion. The term has been deleted.

  1. Add a reference for the ethical criteria you have used – line 207.

R3. This has been done (Page 5, lines 206-207).

  1. Line 223 – should be .82 rather than ,82

R4. Thank you for noticing this mistake. It has been corrected and now appears as .86, as we present the reliability of the original Fraboni scale, instead of other studies that employed it (Page 5, lines 228-229).

  1. Sentence on lines 374-375 is not grammatically correct for the last 5 words – you need to include what is being attributed to old age.

R5. Thank you again for noticing our mistake. The sentence has been rephrased as follows (Page 10, lines 380-382):

Our findings therefore suggest that ageism can indeed be explained by an unconscious attribution of psychological and social vulnerabilities to old age.

  1. Line 381 – suggest you add ‘negatively’ to associated, e.g. associated negatively

R6. We have added this clarification, and rephrased the sentence to avoid any misunderstanding, as it might suggest that we are discussing a statistically negative association between the variables (Page 11, lines 386-388):

It suggests that a subjective experience of vulnerability might unconsciously be associated with negative aspects of the aging process

  1. Line 431 – change ‘predicts’ to ‘is associated with’ as you say below that causality cannot be determined.

R7. This has been done (Page 12, line 441).

We thank you and the reviewers once again for your helpful guidance and constructive feedback which helped to improve our manuscript. We hope that you will find that our revision and our responses in this letter substantively and successfully address each comment and suggestion we received.

Sincerely,

The authors

This manuscript is a resubmission of an earlier submission. The following is a list of the peer review reports and author responses from that submission.

Round 1

Reviewer 1 Report

ijerph-1973231

1.     I think that the definition of ageism is somewhat limited. Not all ageist attitudes are “alteration[s] in feelings, belief…”, and I think that Ayalon et al.’s definition (see https://doi.org/10.1016/j.arr.2019.100919) may be more suitable in this context.

2.     The first paragraph provides a lot of information, and I’m not sure that all of it is relevant to this work. For example, if you are not elaborating on differences between implicit/explicit ageism, it may be better to focus on the physical and psychological consequences of ageism. This can also be applied to the issue of the connection between ageism and abuse/the legal system.

3.     I don’t understand the relevance of the paragraph beginning on line 48. As this issue was not examined directly, this speculative idea should be offered (if deemed necessary) in the discussion.

4.     I think that you raise an important issue in the paragraph beginning in line 54. However, I suggest you elaborate on this issue and describe the association between ageism and self-vulnerability. This is especially relevant when one considers the large age-range you collected. Further elaboration may also assist you in interpreting the interaction you examined.

5.     My main concern with the introduction lies with the assumed direction of the association between the variables (i.e., that vulnerabilities “lead to ageism”, as stated in line 71). Most longitudinal studies report the effect of ageism on well-being, and not vice-versa. This, of course, does not negate the possibility that such psychological vulnerabilities may also be the consequence of ageism, but the cross-sectional nature of this study does not enable the authors to examine this. I believe (in line with my previous comment) that elaborating on how such personal/psychological vulnerabilities may be associated with ageism through Social Identity Theory would greatly reduce this problem.

6.      Line 87-While I agree with this idea when older adults are concerned, I don’t believe that younger adults operate similarly. Please consider rephrasing (or alternatively, elaborate and explain why you believe it to be so).

7.     Line 182- Please refrain from diagnostic language (i.e., “develop ageism”).

8.     Line 288- I’m not sure what MODPROBE is. Please write the model number you used in PROCESS instead.

9.     Please re-write the regression table in accordance with APA requirements (B, SE, confidence interval limits, etc.)

10.  The discussion is written in a manner which suggests causality. Please go through the manuscript and refrain from such notions, which cannot be determined in this study.

Reviewer 2 Report

Dear authors, 

Thank you for the opportunity to read your manuscript about the associations between self-reported psychosocial variables and ageism. The manuscript is easy to read and addresses an important topic with potential practical implications (i.e., ageism). Conceptually, I appreciate that the research is guided by the Social Identity Theory. However, I have some major concerns about the design and data analysis that comprimise the contribution of the study. In that sense, authors indicate that they want "to examine whether ageism is displayed by people in major vulnerable life situations, who, according to the Social Identity Theory, may wish to distinguish themselves from older adults if they recognize the vulnerabilities associated with ageing in their own lives". However, they do not measure identity with any social group (i.e., olders) and do not provide cut-off scores to know whether the reported means and standard deviations of their main variables indicate any vulnerability at all or there is a lack of vulnerability indeed. For example, the PTSD scale seems to range from 20 to 100 and the mean is 37 with a SD of 16, which indicates that most of your participants score between 21 and 53. Similarly, the mean of well-being is 3 points out of a maximun of 4. Therefore, it is difficult to conclude that vulnerability factors in your sample predict more discrimination/prejudice against ageing (i.e., ageism). In addition, as this study did not use any sampling technique, your sample is not representative of the general population. In other words, it seems that high educated women are over-represented in your sample, which questions that your results can be generalized to the general population, which was expected to be one of the major contributions of the study. Other minor issues are: (a) please include the increment in R square to address whether the regression model improves significantly when adding variables (steps); (b) please, due to the cross-sectional nature of your study, avoid causal language and explain how your results can be affected by common method bias (data collected at one moment with self-reported measures); (c) please, in the future, include control measures for "interaction with old people" (as you just assume that your participants interact with them); and (d) please include how similar your participants perceive they are to old people (which is crucial in the social identity theory). 

I hope theses comments and suggestions help you to improve the manuscript and further research in this field.

Round 2

Reviewer 1 Report

I have read the revised manuscript and find that it is much improved. The authors adressed my concerns, and I have no further comments or suggestions.  

Reviewer 2 Report

Dear authors,

Thank you for taking the time to answer previous comments and suggestions. Unfortunately, despite some improvements, the current version has the same flaws: (1) the theoretical framework is not clear. Authors should integrate better the Terror Management Theory and the Social Identity Theory in the introduction (or just follow one theory and explain in detail how ageism emerges and develop according to such theory). Howerver, my suggestion is to frame the paper as a explorative study (seems more inductive than deductive); (2) the lack of measurement of key variables in your study, such as: social identity (or even identity fusion, which can give you a clearer measure of "distance" from old/elderly population depending on participants' age), or the perceptions of the participants about how certain vulnerabilities are associated to old age/ageism. This is a major limitation that prevent me to recommend publication. It is crucial because your storytelling is based on Social Identity Theory but you are assuming that all the mechanisms proposed by Social Identity Theory are playing an important role in explaining the link between age-vulnerabilities-ageism without measuring such mechanisms. In other words, there is a over-interpretation of your results. Overall, your results showed that vulnerabilities are associated to age in a not expected way: the older you are, the less somatization and posttraumatic stress you report (see correlations in Table 1, but positively with community belonging). Indeed, in the hierarchical regression model, it seems that age is negatively correlated to ageism (B = -.18*) and post-traumatic stress is positvely correlated to ageism (B = 17*). Interestingly, there is an interaction effect between age and post-traumatic stress (PTS) on ageism: when there is no PTS, younger people report more ageism; however, when there is moderate-to-high PTS, older people report more ageism. Then, your findings may indicate that getting older is associated with higher PTS and therefore with more ageism (according to Social Identity Theory this can happen because people do not want to be labeled as "old": they want to differentiate themselves from "old" people because they do not want to be associated with the negative characteristics that are associated with "old" people. That can be an explanation but you cannot rule out alternative explanations bacause you did not measure identity with old people, characteristics associated with old people or any other of these potential mechanisms that Social Identity Theory posits... In other words, you may use Social Identity Theory in the discussion as a potential explanation but in the introduction does not make too much sense. Indeed, it does not explain why younger people report more ageism when there is no PTS. Also, it does not explain why the interaction and the only significant beta in the regression is with age and PTS but not with the other "vulnerabilities" and protector factors; why is PTS more salient than the other variables?); and therefore (3) discussion and conclusion should be improved to fit the goal and the findings of the manuscript.